# Revealing Development Trends in Blockchain-Based 5G Network Technologies through Patent Analysis

Fei Gao [1], De-Li Chen [1], Min-Hang Weng [1,*] and Ru-Yuan Yang [2]

[1] School of Information Engineering, Putian University, Putian 351100, China; gaofei8237@163.com (F.G.); deli008@126.com (D.-L.C.)
[2] Graduate Institute of Materials Engineering, National Pingtung University of Science and Technology, Pingtung County 912, Taiwan; ryyang@mail.npust.edu.tw
* Correspondence: hcwweng@gmail.com

**Abstract:** The fifth-generation (5G) network has special communication and security requirements including high reliability, low latency, precise automatic control, secure covert transmission, and evidence traceability. The 5G network combined with blockchain technology just meets this demand, so it is driving a rapidly growing volume of patent applications. This study proposes application scenarios, architecture diagrams, and patent analysis methods for blockchain-based 5G network technologies, beginning with a network architecture using mobile edge computing (MEC) and blockchain as independent platform components to solve MEC load pressure. In the patent analysis, a patent cluster map of blockchain-based 5G networks is proposed to analyze the intersection of technical application fields. The bottleneck period of technological development is presented for leading countries and enterprises in the technological development of blockchain-based 5G network, highlighting relative advantages and disadvantages. Specifically, to extract the core international patent classification (IPC) key technologies and their mutual interrelatedness, we use network topology analysis to establish an IPC network topology diagram through node global and local topology characteristics, thus revealing hotspots of IPC technology research and the characteristics of the technology relationship system. The findings provide a very useful reference for the formulation of government strategy to assist in the implementation and development of blockchain-based 5G network technologies for future smart cities.

**Keywords:** 5G; blockchain; patent analysis; IPC key technology; network analysis

## 1. Introduction

Fifth-generation (5G) mobile communication technologies promise high data throughput rates, low latency, and massive access, allowing for the rollout of user-centric 5G services to meet the growing needs and expectations of wireless internet users [1–3]. 5G provides data computing and high-speed transmission for massive connections to support heterogeneous Internet of Things (IoT) devices. However, the deployment of 5G communication networks raises new requirements and challenges. Blockchain allows many different devices to coexist and interoperate in a transparent, safe, and friendly manner [4]. This new infrastructure paradigm can help resolve issues related to device registration and authentication, energy security transactions, and data traceability, thus compensating for some critical shortcomings in 5G technology [5]. The use of blockchain can help promote the efficient development of 5G applications, optimizing the underlying 5G network security communication technology [6,7]. Blockchain provides a level of security unmatched by existing centralized infrastructure [8–10]. The use of 5G technology can effectively improve the data processing rate of the blockchain, reducing user's response times while increasing transaction frequencies [11,12].

These developments have driven a rapid growth in the number of patent applications related to the integration of blockchain into 5G networks, raising several urgent issues:

1. How are networks deployed based on the integration of 5G and blockchain technology structured?
2. Which countries/companies/applicants/international patent classifications (IPCs) are active in this patent application domain?
3. Has a technology cluster or technology distribution formed in this patent application domain?
4. How can more important patents be identified from among existing patent applications?
5. How can future technical development directions of patent application layout strategies be identified from existing applications?

To address these questions, this paper uses a novel patent analysis method and focuses on the following contributions:

We analyze the integrated architecture of the blockchain-based 5G network and propose a network architecture topology that separates the mobile edge computing (MEC) from the blockchain system and describe the advantages of this topology. We also analyze and summarize the network deployment characteristics of the blockchain-based 5G network in some patent documents.

- We develop a patent research framework and research methods to analyze the technology life cycle and bottleneck period. The top five countries and the top seven patent applicants in this domain are ranked in terms of the number of patents and R&D ability.
- We categorize the patents according to keywords and analyze the intersection of 5G and blockchain technologies in the application domain to provide a better understanding of the patented technologies. Patents in the cross nodes of the technology cluster map are selected for detailed analysis.
- We further adopt the network topology to map the IPC, with each node of the network topology representing an IPC technology. The topology diagram describes the relationship network and relationship strength between IPCs. The global and local topological properties of nodes in the network topology are used to evaluate node importance, which is then used to identify core IPC technologies.
- Finally, we summarize the research significance and some specific suggestions to avoid problems in the implementation and coverage of blockchain-based 5G network, which has significant implications for practice.

Currently, patent analysis on 5G and blockchain only considers these domains in isolation [13,14], but does it in tandem, particularly for blockchain-based 5G networks. The motivation of the present study analyzes patents of technologies related to blockchain-based 5G networks, classifying relevant patent information, and evaluating key technology development trends. In addition, this study uses a network topology graph to analyze the topological correlation between IPC node technologies and the IPC core technology.

The organization of this paper is described as follows: Section 1 introduces the research background and motivation of this study concerning the 5G network and blockchain; Section 2 reviews the literature about the relevant technologies to be mentioned in the research, including the significance of 5G network, blockchain, patents, and patent analysis; Section 3 introduces the research methodology and the research process; Section 4 shows the analyzed results and the findings of the entire research; and Section 5 summarizes recommendations of the patent strategy.

## 2. Literature Review

### 2.1. Research Status

5G network and blockchain technology can be integrated to develop applications serving domains including trade and finance [15], smart cities [16], smart hospitals [17], supply chain management [18], Internet of Vehicles [19], energy internet [20], digital asset management [21,22], and Industrial Internet [23].

At present, countries around the world are working at full speed to implement 5G and blockchain technologies in practical applications On November 26, 2019, Cisco received approval for a patent that may have far-reaching implications for blockchain-based smartphone user identification, introducing a system that can be used in 5G networks to verify user identities and facilitate online payment services [24]. South Korea's Sungkyunkwan University has proposed a real-time atmospheric pollution index measurement platform based on 5G wireless networks and blockchain [25]. Also in South Korea, a team at Sejong University has designed a 5G-based edge vehicle computing model and a blockchain-based secure event-driven message (EDM) protocol to provide EDM auditability [26]. Singapore's Nanyang Technological University uses 5G networks to enable vehicles, sensors, and other countless devices to interact and share data on cellular networks, overcoming LTE shortcomings through the blockchain, and resolving handover security problems [27].

The specific technical advantages of integrating 5G network with blockchain technology mainly include the following.

- Data capitalization methods can be based on blockchain smart contracts and can thus be used to control IoT devices through blockchain [28,29]. Some new studies on smart-contract-based data commodity transactions for Industrial IoT have been proposed [30]. Terminal equipment can be scheduled and accessed controlled by blockchain platforms [31,32]. Blockchain enables IoT devices to implement authentication technologies over fiber or 5G wireless networks [33].
- Each blockchain network can operate independently in different slices. Slicing technology can be combined with resource mapping of virtualization technology or network function virtualization (NFV) in 5G networks, effectively improving the utilization rate of 5G network spectrum resources for blockchain data [34–36].
- Blockchain can be integrated into mobile edge computing (MEC) applications in 5G networks, such as distributed resource allocation [37]. Predefined rules are implemented using blockchain smart contracts on mobile edge computing networks or MECs to effectively reduce network congestion and maximize the user's service quality experience of terminal node users. [38].

## 2.2. Network Architecture Diagram of Blockchain-Based 5G Network

Blockchain requires considerable computing power resources. In 5G networks, MEC provides optimal support for blockchain development [39–41]. Figure 1 shows a network architecture platform based on 5G wireless networks and blockchain. The platform collects data in real time through the IoT, and all IoTs devices are connected to sensors through 5G networks. The platform is divided into four layers, namely (1) device perception layer, (2) data processing layer, (3) network core layer, and (4) data preservation layer. The device perception layer provides modules to track device positioning, energy status, device abnormality, radio frequency identification (RFID), sensors, fingerprint identification, and face recognition. Through the data processing layer, the 5G mobile edge subsystem MECs provide distributed processing capability to collect and accelerate data processing and analysis. It shares the task loading with core cloud computing resources and effectively improves cloud responsiveness. This layer is located in the edge cloud, and the MEC 5G network edge subsystem accelerates the processing and analysis of data collected through distributed processing capabilities while also improving cloud computing responsiveness. The network core layer provides high reliability, high security, high speed, wide coverage, low latency, and data redundancy. The data preservation layer matches the contract according to the agreed trading rules through the blockchain platform and then confirms the transaction. Finally, the data and events are uploaded to the blockchain platform, which can also conduct security scheduling for terminal devices connected to the 5G network. Thus, the blockchain serves a security function for 5G networks.

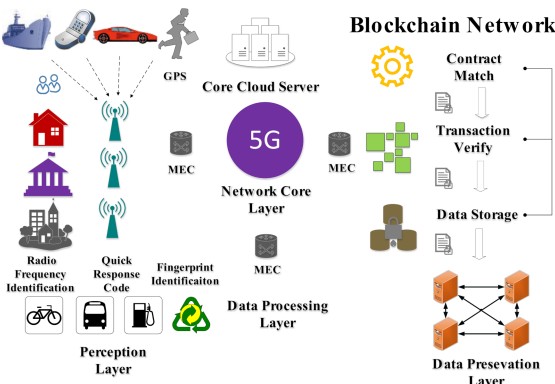

**Figure 1.** Network architecture platform based on a fifth-generation (5G) wireless network and blockchain.

Figure 2 shows the integration architecture of MEC and blockchain, based on a 5G and blockchain integration development and application white paper released by a team in China focused on blockchain and MEC technology [42]. The architecture is divided into three layers: (1) The bottom MEC IaaS (Infrastructure as a Service) layer is responsible for the allocation and scheduling of computing, storage, and network resources. It can also provide server resources for external blockchain systems and schedules 5G base station fragments to allocate 5G resources. Each fragment can be independently assigned to the blockchain network. (2) The MEC platform PaaS (Platform as a Service) layer provides network and professional capacity, while the blockchain platform provides the core blockchain support functions including block storage, smart contracts, and consensus, which enrich the MEC capabilities. The capability open subsystem in the MEC platform can enable services for upper-level carrier-level APPs and third-party APPs. At the resource level, the resources required by the blockchain platform are subject to unified scheduling and allocation. (3) The SaaS (Software as a Service) layer includes the entire system's application service capability.

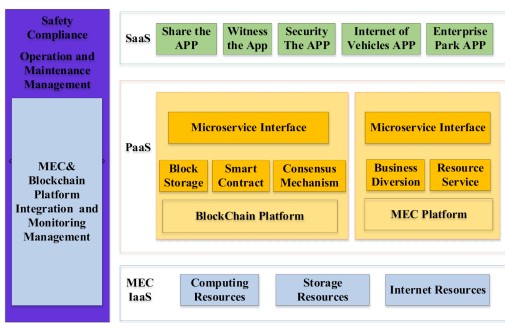

**Figure 2.** Network architecture platform based on integrated mobile edge computing (MEC) and blockchain platform components.

Figure 3 shows a 5G network deployment architecture with MEC and blockchain as independent platforms. This scenario is similar to the deployment of traditional blockchain systems but requires unique devices to access the business channel to provide seamless 5G access. The modem processes the data transmitted between the blockchain platform and 5G network, analyzes the modulated data, and provides communication between the MEC and the blockchain platform. The MEC provides a routing subsystem, terminal device positioning, and sensor data feedback for blockchain. Blockchain provides an authentication mechanism to help MEC verify terminal equipment and save important events on the chain. Through the blockchain platform control, the MEC can open blockchain query permissions for authorized users.

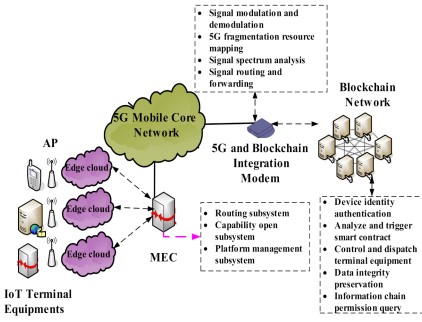

**Figure 3.** Network architecture platform based on MEC and blockchain as independent platform components.

Table 1 shows the technical characteristics, deployment features, and application areas for the integration of blockchain technology and MEC or edge computing [43–52].

**Table 1.** Integration of blockchain and MEC or edge computing technology.

| Technical Description | MEC or Mobile Edge Computing Technology | Blockchain Technology | Architecture Level/Deployment Features | Application Field |
|---|---|---|---|---|
| A blockchain-based trusted data management scheme in edge computing [43] | MEC data storage/data management/data encryption security | Smart contract/identity authentication | MEC IaaS layer/integration deployment | Security and privacy protection of edge device data |
| A novel distributedblockchain-based trusted MEC collaboration [44] | MEC topology privacy protection | Data tamper-proof/identity authentication | MEC IaaS layer/independent deployment | Trust and privacy protection |
| Bringing blockchain technique into fog environment so as to verify each fog server's authenticity and propose a blockchain-based offloading approach [45] | Data offloading of edge terminal devices or MEC | Identity authentication | MEC IaaS layer/independent or integration deployment | Reducing mobile devices or MEC workload |
| Energy trading framework based on blockchain [46] | Generating unique consensus machine for each electric vehicle system | Secure energy trading | PaaS layer/ integration deployment | Vehicle safe energy trading |
| A vehicular blockchain-based secure and efficient GPS positioning error evolution sharing framework [47] | A DNN-based error correction algorithm that runs on the edge server | Smart contract for data storage and sharing | SaaS layer/ integration deployment | GPS positioning in vehicular networks |
| Access control of electronic health record data [48] | Storing EHR data and collaboration with blockchain-based accessing control logs | Managing identity and accessing control policies | PaaS layer/ independent or integration deployment | Electronic health records |
| The interaction between the cloud/fog providers and the miners in a proof of work-based blockchain network [49] | Resource management and network security | Proof of work algorithm | MEC IaaS layer/integration deployment | Load migration and reducing computing resource requirements |
| Design and prototype an edge-IoT framework based on blockchain and smart contract [50] | Resource allocation | Smart contacts for controlling IoT devices | PaaS layer/integration deployment | IoT behavior/resource scheduling standardization |

**Table 1.** *Cont.*

| Technical Description | MEC or Mobile Edge Computing Technology | Blockchain Technology | Architecture Level/Deployment Features | Application Field |
|---|---|---|---|---|
| Blockchain-based mobile edge computing share system [51] | Processing user terminal requests | Data security sharing | MEC IaaS layer/ independent or integration deployment | Wireless communication bandwidth enhancement for smart cities |
| Utilizing drones combined with blockchain technology to ensure safety during data collection [52] | Data storage and verification | Data tamper-proofing | MEC IaaS layer /integration deployment | Transmission and security of drone data |

*2.3. Patent Analysis*

Given increasingly fierce global technological competition, companies increasingly rely on patent strategy research and patent analysis. Analyzing, processing, and combining patent information creates competitive intelligence that can be subjected to statistical prediction techniques to provide companies with a useful reference for the development of technologies, products, and services. Statistical analysis methods can be used to evaluate and predict international patents.

Not only is patent analysis is a prerequisite to patent-based competitiveness, but it also provides companies with useful information for companies to develop their technological strategies, evaluate competitors, and anticipate market trends. Patent analysis is a uniquely practical analysis method for developing enterprise strategy and competition analysis, generating visual statistical reports to illustrate technical activities, and using life cycle models to predict patent development trends [53–55]. In this study, we analyze the development of 5G and blockchain through patent analysis technology. Moreover, we use network topology characteristics to reveal patent development trends and key IPC patent analysis technologies.

**3. Research Architecture**

*3.1. Research Methodology*

This patent analysis was conducted using IPTECH software, a comprehensive patent analysis platform with global patent search and analysis tools developed by Taiwan LianYing Technology Co., Ltd. in 2003. This analysis database comes from the patent database website of various countries. The database provides searchability of patents filed in the People's Republic of China, the Republic of China (Taiwan), the United States, the European Union, and Japan, along with WO-PCT and other patent databases, making it one of the most competitive patent database search service platforms available. Patent information is conducted in terms of number of patents, life cycle, number of IPCs, IPC correlation strength, development trends of key technology, etc.

*3.2. Research Process*

Figure 4 shows the patent analysis process for the blockchain-based 5G network. In the first step, we study the background significance of the subject and specify the research motivation. In the second step, we then identify the paper's research themes, where research topics of interest are determined based on the research significance of these themes. In the third step, we study the development history, related research status, and technical characteristics of the research subject from the periodical and patent literature, to select the desired keywords. In the fourth step, we conduct keywords as search items to identify the relevant patents. In the fifth step, we screen the search results using text exploration and patent focus reading methods and remove the irrelevant patents. In the sixth step, the relevant patents are then classified into management analysis and technical

analysis through the built-in statistical analysis of the IPTECH software. The management analysis mainly includes annual trend analysis of patent activity, the patent life cycle, IPC development trends of the patent technology, and inventor analysis. Management analysis of patents is conducted from multiple perspectives to provide a macro understanding of development trends and the overall picture of specific patents. The technical analysis mainly includes a technology cluster map.

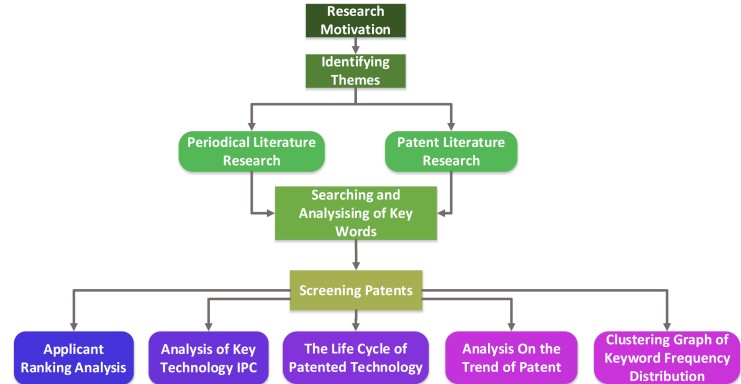

**Figure 4.** Patent analysis process for blockchain-based 5G network.

Moreover, many patent charts only show the number of patents in the primary IPC but fail to show the correlation between the key patent technologies and the significance of the IPC. However, measuring the importance of IPC technologies is not only related to the number of primary IPCs but also to the closeness between them. Therefore, the node-to-node network topology is further used to evaluate the correlation and importance between IPCs based on the characteristics of the topology graph nodes.

## 4. Results and Discussion

### 4.1. Analysis Of 5G-Related Patents

Research technology terms related to 5G network typically include array antenna, multi-carrier, multiplexing technology, super-dense network, software-defined, self-defense networks, optics, semiconductors, nanostructures, virtualization, slice, cloud computing, edge computing, software-defined network, network virtualization, network slice, and (device-to-device communication) D2D network.

Table 2 presents a statistical summary of the number of patents in IPTECH based on the name, abstract, patent application scope, and patent description of 5G combined with the abovementioned technical fields. Percentage of patents is used to evaluate and compare the 5G patent growth rate in different technical fields, where the percentage of patents is the sum of the number of patents (approved plus public patents) over the past three years divided by the sum of the number of patents over the past six years. The domain with the largest percentage is 5G with multi-carrier technology. In contrast, 5G with blockchain technology has the lowest percentage overall but the highest over the past three years, indicating this domain is an emerging application field with considerable future research space. It is clearly found that as technological bottlenecks are resolved, the next few years should be a period of rapid growth in the number of patents related to blockchain-based 5G networks.

**Table 2.** Patent Statistics Retrieved By 5G Technology and Related Fields.

| TAP AND KEYWORD | 2014–2020/ Approved | 2014–2020/ Public | 2017–2020/ Approved | 2017–2020/ Public | Percentage of Patents |
|---|---|---|---|---|---|
| TACD:(5G) AND TACD:(BLOCKCHAIN) | 123 | 1130 | 122 | 1128 | 99.76% |
| TACD:(5G) AND TACD:(ARRAY ANTENNA) | 28,957 | 98,557 | 17,903 | 76,425 | 73.97% |
| TACD:(5G) AND TACD:(MULTI-CARRIER) | 581,198 | 1,218,001 | 142,299 | 428,808 | 31.74% |
| TACD:(5G) AND TACD: (FULL DUPLEXREUSE) | 24,326 | 67,509 | 11,282 | 41,358 | 57.32% |
| TACD:(5G) AND TACD: (SUPERDENSE NETWORK) | 28,914 | 114,539 | 20,175 | 98,472 | 82.71% |
| TACD:(5G) AND TACD: (SOFTWARE DEFINED) | 56,335 | 147,798 | 24,374 | 85,411 | 53.78% |
| TACD:(5G) AND TACD:(SDN) | 893 | 3562 | 754 | 3185 | 88.42% |
| TACD:(5G) AND TACD:(OPTICS) | 4024 | 8990 | 1677 | 4214 | 45.26% |
| TACD:(5G) AND TACD:(SEMICONDUCTORS) | 2073 | 5024 | 765 | 2562 | 46.88% |
| TACD:(5G) AND TACD:(NANOSTRUCTURES) | 397 | 1055 | 200 | 512 | 49.03% |
| TACD:(5G) AND TACD:(VIRTUALIZATION) | 1340 | 6722 | 1265 | 6403 | 95.11% |
| TACD:(5G) AND TACD:(SLICE) | 2236 | 9416 | 1259 | 7763 | 77.43% |
| TACD:(5G) AND TACD:(MEC) | 1183 | 3851 | 566 | 2598 | 62.85% |
| TACD:(5G) AND TACD:(D2D) | 2529 | 22,882 | 2367 | 20,337 | 89.35% |

(T = Name, A = Abstract, C = Patent Scope, D = Patent Description).

### 4.2. Management Analysis

Next, we conducted a patent analysis of blockchain-based 5G network according to the search criteria "TACD: (5G) and TACD: (Blockchain)" for management analysis. A total of 1253 patents related to blockchain based on 5G network technology were identified from January 2014 to April 2020.

Figure 5 shows an annual trend analysis for the number of blockchain-based 5G network patents, showing a continual rise from 2014 to 2018 and a decline since 2019.

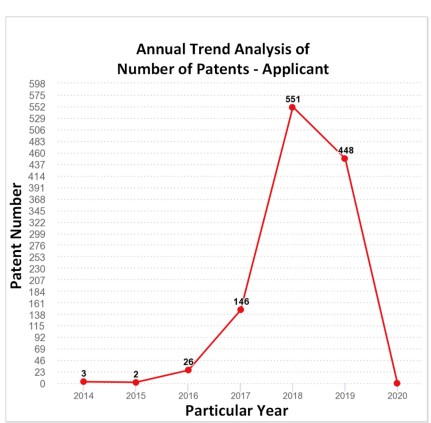

**Figure 5.** Blockchain-based 5G network patent application number (January 2014 to April 2020).

Figure 6 shows the life cycle of blockchain-based 5G network patents, showing the relationship between the number of patent applications and the number of patentees from 2014 to 2019. In 2014, the technology was in its infancy, while the years 2015 to 2017 presented a growth period. By 2018, the technology had entered a mature stage, with

patent approvals peaking during this period. Since 2019, the number of patentees has increased as the overall number of patent grants has fallen, indicating the emergence of a technology bottleneck, requiring further innovation and breakthroughs to sustain further robust development.

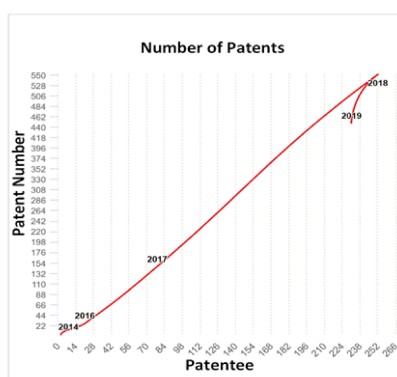

**Figure 6.** Blockchain-based 5G network patent life cycle (January 2014 to December 2019).

Figure 7 summarizes statistics for patent applications in multiple countries from 2014 to 2020, identifying the top five countries or organizations of origin. The change trend of the curve indicates a peak of patent research in 2018, followed by a decline in 2019. As previously mentioned, 2019 presents a bottleneck period for the development of 5G and blockchain. The distribution map of patents shows that the United States (684 patents), China (114), and Germany (41) are the most competitive in this space, followed by Finland (23) and Great Britain (16). Among these countries, the United States, China, and Great Britain attach considerable emphasis to blockchain research and application development, while Germany is actively developing its blockchain industry, and China and Finland have successively formulated comprehensive regulatory frameworks for blockchain supervision.

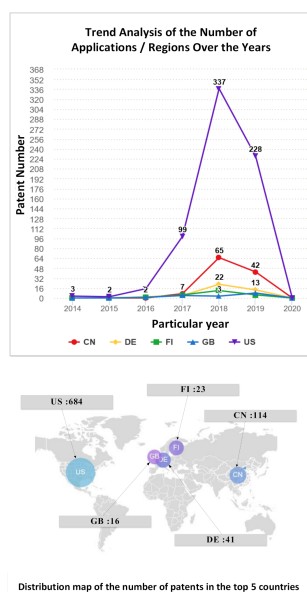

**Figure 7.** Top five countries of origin of blockchain-based 5G network patent application (January 2014 to April 2020).

At the corporate entity level, United States corporations accounted for five of the top seven applicants for blockchain-based 5G-network-related patents, including the intellectual property arms of Cisco, Intel, AT&T, Bank of America, and Microsoft, with patents reflecting deeper integration of technologies including MEC, slice, IoT, and NFV. Only one

Chinese company made the top seven: Alibaba, which is the world's leading patent holder for blockchain-related technologies and has also applied for many patents related to the bottom layer and application fields of blockchain in support of the 5G communication protocol standard. However, Alibaba's patent coverage for the integration of 5G applications with IoT and blockchain technologies is still somewhat superficial. In 7th place is Finland's Nokia, which released a blockchain-based sensor system to help smart city development, creating a proof of concept in collaboration with British Telecom (BT). Germany's Siemens AG is also actively pursuing patents in the blockchain-based 5G domain, while Great Britain's Arm LTD is focused on integrating AI, 5G, and IoT.

Table 3 shows search results for blockchain-based 5G network technology patents based on a range of different search conditions. According to the third record in Table 3, 393 patents were screened out from the 542 patents through patent exploration, reading, and merging. These patents were then subjected to technical analysis for blockchain-based 5G networks.

**Table 3.** IPTECH search records.

| Keyword Search | Number of Patents Searched |
|---|---|
| TACD:(5G) AND TACD:(BLOCKCHAIN) | 1253 |
| TAC:(5G) AND TACD:(BLOCKCHAIN) | 34 |
| TACD:(5G) AND TAC:(BLOCKCHAIN) | 542 |
| TAC:(5G) AND TAC:(BLOCKCHAIN) | 20 |

Table 4 further shows the top seven patent applicants for the corresponding IPC technology, the corresponding patent application country for the PCT system, and the resulting number of patents. All applicants are mature global enterprises. These organizations of the patent rights are internationally renowned enterprises. They are led by Alibaba Group Holding Limited, which is ranked first with 89 patents. Next is Intel Corporation, which is ranked second with 24 patents. Nokia Corporation is ranked third, with 13 patents. The fourth and fifth places are Bank of America and Corporation and Microsoft Technology Licensing, LLC, respectively, which both have 12 patents. The patent strategies of these enterprises mainly focus on the transmission of digital information (H04L), data processing systems or methods (G06Q), wireless communication networks (H04W), and electrical digital data processing (G06F).

**Table 4.** Ranking of the top seven patentees.

| Patentee | Patent Number | Primary IPC | Patent Applications (Approved, Published) |
|---|---|---|---|
| Alibaba Group Holding Limited | 89 | G06Q, H04L, G06F | US (80,22), EU (3,2),PCT (37,0) |
| Intel Corporation | 24 | H04L, G06Q, H04W | US (15,0),EU (1,0), PCT (8,0) |
| Nokia Corporation | 13 | H04L, H04W, G06Q | US (5,0), EU (7,0), PCT (9,0) |
| Bank of America Corporation | 12 | H04L, G06Q | US (12,3) |
| Microsoft Technology Licensing, LLC | 12 | H04L, G06Q, G06F | US (9,3), PCT (6,0) |
| Cisco Technology, INC | 10 | H04W, H04L | US (6,3), PCT (4,0) |
| AT&T Intellectual Property I,L.P. | 10 | H04L, G06F, G06Q | US (10,2) |

Figure 8 shows the number of patents applied by patentee around 2018. Prior to 2018, Intel was the leading patent applicant, with a focus on invention patents, suggesting the blockchain-based 5G network domain was still in the technological R&D stage and had not yet entered the product model development stage.

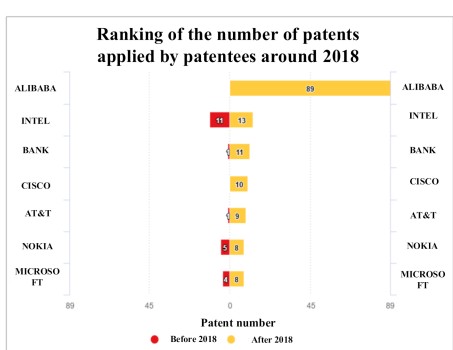

**Figure 8.** Number of patents applied by patentee around 2018.

Table 5 ranks the relative R&D capacity of the various patentees, showing superior capacity from Alibaba, Intel, and Nokia. The efforts of Intel and Bank of America mainly focused on the virtualization of integrating blockchain and 5G in IoT applications. A company with a higher number of patent applications might have better research and development strength in industry technology, a company with a higher number of other citations may have the core patent in the technical field, a company with more patent self-citations might have a R&D direction with more important technology branch of the field, and a company with a higher number of inventors might pay more attention to patented inventions in this field. Patent Age is the average patent age in the company, and a company with a younger patent age might enjoy a longer technological advantage in the field. Active year means that the patentee has a long-term patent export activity in the technology. The parameters corresponding to this formula are the default parameters set in the IPTECH patent database for patent strength evaluation. While Alibaba led the field in terms of total number of patent applications and R&D intensity, its performance in deep integration of 5G and blockchain was relatively lower. The weakness mentioned here means that most of Alibaba's patents in the development of blockchain conform to the interface standards of 5G networks, but it is not as good as Cisco in terms of technological integration, as can be seen from Table 8. Alibaba needs further research and development in terms of the degree of deep integration of 5G and blockchain technology. Cisco ranked last in terms of the absolute number of patent applications but had focused on quality over quantity (see Table 8), using blockchain technology to combine 5G network slices, MEC, micro-services, and network functions virtualization technology.

### 4.3. Analysis of IPC and Technology Cluster Map

Figure 9 shows a technology cluster map for blockchain-based 5G network patent applications. This information is typically concealed and difficult to obtain directly from patents. The cluster keywords are shown in the image, and the numbers in parentheses indicate the number of patents corresponding to each keyword. Each yellow dot represents one patent, and different colored lines represent different keywords associated with the patent. The more links to a dot, the more keywords tied to the patent and the more extensive its content. The technology direction of the patents is found to involve multiple fields, including computer-implemented methods, smart contracts, distributed ledgers, computing devices, blockchain members, Internet of Things, blockchain authorization, and mobile devices. The greatest number of patents (86) is related to computer implementation methods, followed by distributed ledgers (53), computing devices (51), and smart contracts (50).

**Table 5.** Ranking of R&D capability of patentees.

| Patentee | Number of Patent Application $\theta_1$ | Others Citations $\theta_2$ | Self Citations $\theta_3$ | Inventor Number $\theta_4$ | Patent Age $\theta_5$ | Activity Year $\theta_6$ | Relative Capability |
|---|---|---|---|---|---|---|---|
| Alibaba Group Holding Limited | 89 | 0 | 0 | 39 | 1 | 2 | 100% |
| Intel Corporation | 24 | 0 | 0 | 58 | 2 | 3 | 36% |
| Nokia Corporation | 13 | 0 | 0 | 29 | 2 | 4 | 19% |
| Cisco Technology, INC | 10 | 0 | 0 | 23 | 2 | 2 | 15% |
| AT&T Intellectual Property ILP | 10 | 0 | 0 | 26 | 2 | 3 | 15% |
| Microsoft Technology Licensing, LLC | 12 | 0 | 0 | 10 | 2 | 3 | 14% |
| Bank of America Corporation | 12 | 0 | 0 | 7 | 2 | 3 | 13% |
| Weight Coefficient | 5 | 2 | 1 | 1 | −1 | 0 | R&D |

$$(\mathbf{R\&D} = 5\theta_1 + 2\theta_2 + \theta_3 + \theta_4 - \theta_5 + 0\theta_6).$$

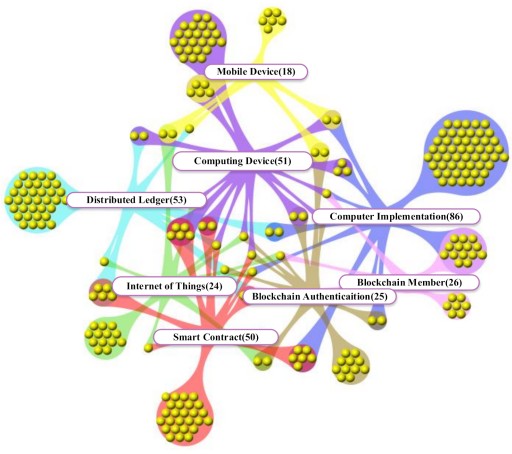

**Figure 9.** Technology cluster map of blockchain-based 5G network technology patent applications.

Moreover, by analyzing the number of nodes at the intersection of the different colored lines, we found significant correlations between certain patent fields. Computer implementation methods are closely related to blockchain members and smart contracts. Smart contracts are closely related to distributed ledgers, computer implementation methods, and computing devices. Blockchain authorization is related to every field. Internet of Things is closely related to mobile devices, smart contracts, and blockchain authorization. Taken together, 5G mobile devices are closely related to Internet of Things access and blockchain authorization. At the same time, the realization of the blockchain requires hardware and software support (i.e., computer equipment) that can support the blockchain architecture and complete the protocol operation. Smart contracts rely on computer equipment to complete the construction of the distributed blockchain ledger system for authentication, control, and scheduling of IoT devices. Most patents comply with the 5G wireless interface standard in the patent application scope and patent specifications, and most smart terminal devices support the 5G mobile communication network interface standard.

*4.4. Key Technology Analysis of IPC Network Topology*

Studying the importance of nodes and their interrelationships in complex networks has important research value [56–58]. Topological characteristics can be used to identify

key nodes and then their correlations. In this study, 30 of the 393 patents are selected to evaluate the key technologies of fourth-order IPCs from the perspective of network topology characteristics. Each IPC is presented as a node in the topology graph to analyze the adjacency relationship between the fourth-order IPCs. This accentuates the links between the network nodes, and the strength of these links also reflects the structural relationship between the IPC node technologies. The topological properties of network nodes are used to obtain the ranking of the importance of key IPC node technologies. To clearly characterize the importance of the local and global topology of network nodes, this paper uses the following node topology characteristics: (1) degree, (2) hits, (3) pagerank, (4) node betweenness, (5) eccentric distance, and (6) clustering coefficient. Each key technology is ranked based on the importance of each characteristic index, and the harmonic mean is used to obtain their comprehensive ranking sequence in the network topology.

1.    Degree

Degree is the simplest but most important property of a node in a complex network. A node's degree is defined by the number of nodes to which it is connected. Intuitively, the greater a node's degree, the more important it is in the topology. For directed graphs, a node's degree is divided into in-degree and out-degree. The in-degree of a node is the number of nodes pointing to the node, and the out-degree of a node is the number of nodes to which the node points. Here, a node's degree is equal to the sum of its in-degree and out-degree. The in-degree is the number of primary IPCs related to the node. The out-degree refers to a certain node being the primary IPC, and a node's out-degree refers to the number of IPC nodes involved in the same patent as the node in question.

2.    Hits

In the hit algorithm, the directed graphs is divided into authority nodes and hub nodes. Authority nodes refer to the high-quality nodes related to a certain domain, while hub nodes are those nodes that contain many links to high-quality authority nodes. The basic idea is to use the reference chain between nodes to mine useful information. For example, the primary IPC in a single patent is the authority node, and the hub is the other IPC node. This method is used to study the linkages between the primary IPC node and the other IPC node to construct a directed topology graph.

3.    Pagerank

The pagerank algorithm is based on directed graphs [59]. It freely traverses the graph to calculate a probability distribution representing the probability of a user randomly clicking on a link to visit a certain web page, and this probability is the node's importance. The idea of pagerank is that if a node is linked to by other nodes, it means that this node is more important, and the pagerank value will be relatively high. We use the pagerank algorithm to introduce the concept of network link value to the evaluation of the importance of complex network nodes. The higher the network link value of the node, the more important the node is.

4..    Node betweenness

Node betweenness is defined as the ratio of the number of shortest paths passing through the node to the total number of shortest paths of all nodes in the topology [60]. The larger the node betweenness value, the more paths pass through the node, and the more important the node is in the topology. The node betweenness reflects the importance of the node as a bridge. It is an important global topological property.

5.    Eccentric distance

The node's eccentric distance is the maximum value of the shortest distance from the node to all other nodes, according to the definition of graph theory, The eccentric distance can represent a node's relative position from the topological center of gravity, and the node's importance can be evaluated from the global characteristics of the topological structure. The closer the node is to the center of gravity, the more important it is.

## 6. Clustering coefficient

The clustering coefficient of the network plays an important role in the link formation [61]. The greater the aggregation of nodes, the easier it is for them to transmit and exchange through the network. Therefore, the higher the local clustering coefficient of a node, the higher the probability that the node is in a dense sub-cluster and the more representative it is. This study uses the local clustering coefficient as the topological characteristics of the nodes.

The harmonic average of each attribute ranking of each node was calculated, and the importance order of each node in the network topology was obtained.

Figure 10 shows the network topology structure diagram of the IPC technology with fourth-order structure. The nodes represent each IPC technology, and the correlation strength between nodes is distinguished by the thickness of the line, where a thicker line denotes greater relevance. The size of the node indicates the importance of IPC technology in the IPC relational network. In other words, the importance of the node in the topology diagram reflects the greater importance of the structural hole of the node in the network. A node with structural hole characteristics indicates it occupies a more important position in the network, while the dark-colored node indicates that the IPC is the primary IPC patent. The graph reflects the relationship strength between IPC technologies. Among the 30 patents, strong correlations were found between H04L 9, G06Q 20, and G06F 21; between H04L 29 and H04W 12; and between H04L 9, G05D 1, and G07C 5.

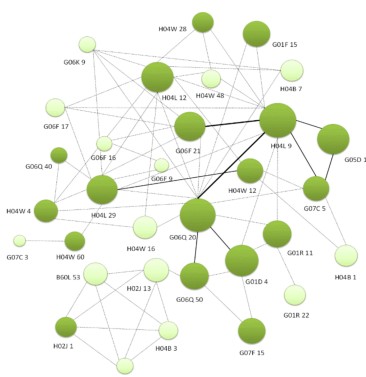

**Figure 10.** Network topology structure diagram of international patent classification (IPC) technology with fourth-order structure.

Next, the patent's key IPC technology was further analyzed. If the number of IPCs or the number of primary IPCs is larger, the proportion of research content in this technology field is larger. Therefore, this study takes the number of IPCs and primary IPCs as measurement indexes of patent technology. We finally selected 10 current major IPC key technologies, ranked in descending order H04L 9, G06Q 20, H04L 29, G06F 21, H04L 12, G06Q 50, H04W 12, G01D 4, G05D 1, and G07C 5.

Table 6 shows the importance ranking of the IPC key technologies. The harmonic mean is the comprehensive ranking of each IPC in a different index recommendation order. The smaller the value, the higher the ranking. Table 7 shows the ranking of the number of IPCs and the number of primary IPCs. Among them, the top three IPC technologies occupy a high proportion in network topology characteristics, the number of IPCs, and the number of primary IPCs.

**Table 6.** Importance ranking of IPC key technologies.

| IPC | Degree | Hits | Pagerank | Node Betweenness | Eccentric Distance | Clustering Coefficient | Mean |
|-----|--------|------|----------|------------------|--------------------|------------------------|------|
| H04L 9 | 12 | 1 | 1.1117 | 137 | 7 | 0.5366 | 2.0269 |
| G06Q 20 | 13 | 0.1178 | 1.0323 | 365.6 | 6 | 0.5078 | 2.0282 |
| H04L 29 | 11 | 0.1545 | 0.9435 | 268.6 | 7 | 0.3620 | 5.0144 |
| G06F 21 | 6 | 0.1182 | 0.6945 | 33.50 | 7 | 0.7779 | 6.1714 |
| H04L 12 | 11 | 0.2321 | 0.7660 | 178.2 | 8 | 0.4097 | 5.4569 |
| G06Q 50 | 4 | 0.1311 | 0.6049 | 260 | 7 | 0.7701 | 6.1586 |
| H04W 12 | 4 | 0.2401 | 0.5906 | 34.66 | 8 | 0.4070 | 8.9255 |
| G01D 4 | 5 | 0.3481 | 0.6402 | 129 | 5 | 1.0952 | 3.1059 |
| G05D 1 | 2 | 0 | 0.5310 | 0 | 10 | 3.6342 | 4.9633 |
| G07C 5 | 5 | 0.1760 | 0.6716 | 59.50 | 7 | 0.7519 | 8.0769 |

**Table 7.** Frequency statistics and application characteristics of IPC.

| IPC. | Frequency of Appearance | Primary IPC Appearance | Application Characteristics |
|------|-------------------------|------------------------|----------------------------|
| H04L 9 | 250 | 85 | Confidential or secure communication device |
| G06Q 20 | 147 | 83 | Payment architectures, schemes, or protocols |
| H04L 29 | 175 | 72 | Arrangements, apparatus, circuits, or systems |
| G06F 21 | 90 | 40 | Security arrangements for protecting computers, components, programs, or data against unauthorized activity |
| H04L 12 | 29 | 11 | Data switching networks |
| G06Q 50 | 31 | 9 | Systems or methods specially adapted for a specific business sector |
| H04W 12 | 48 | 12 | Security arrangements, authentication |
| G01D 4 | 4 | 1 | Tariff metering apparatus |
| G05D 1 | 4 | 3 | Control of position, course, altitude, or attitude of land, water, air, or space vehicles |
| G07C 5 | 4 | 1 | Registering or indicating the working of vehicles |

For the IPC classification of the fourth-order structure, blockchain-based 5G network technologies are currently mainly concentrated in confidential or secure communications equipment (H04L 9), payment architecture solutions or protocols (G06Q 20), devices or circuit system (H04L 29), and infringement prevention and data exchange (G06F 21). An IPC with a three-order structure is mainly concentrated in the fields of H04L (transmission of digital information), G06Q (data processing system or method), G06F (electric digital data processing), and H04W (wireless communication networks).

Table 8 analyzes the patents with high citation rates for blockchain-based 5G networks or patents with more technical fields in the past three years. It lists the main research features, IPC technologies, and type of application.

**Table 8.** Studies of the primary patented technology.

| Sn | International Patent Name | Patentee | Types of Applications | IPC |
|----|---------------------------|----------|------------------------|-----|
| 1 | Securing communications for roaming user equipment using a native blockchain platform | Cisco Technology, Inc., San Jose, CA, USA | Identity authentication | H04L 29, H04W 12 H04W 60, G06Q 20 |
| 2 | Blockchain-based auditing, instantiation, and maintenance of 5G network slices | Cisco Technology, Inc, West Tasman Drive, SanJose, CA, USA | Authentication/credit audit /information sharing | G06F 21 |
| 3 | Systems and methods for collaborative road user safety | Alibaba Group Holding Limited, Hangzhou, China | Collaborative safety devices belonging to road users | H04L 9, G08G 1, G06F 16, H04W 4 |
| 4 | Intelligent electric meter system capable of carbon emission reduction calculation | Hepu Technology Development, CUI, Beijing, China | Power/smart grid/carbon emissions | G01R 11, G01R 22 |
| 5 | Method for controlling platooning and autonomous vehicle based on blockchain | Lg Electronics Inc., Seoul, Keara | Internet of vehicles control and scheduling | H04L 9, G05D 1, G07C 5 |
| 6 | Multi-access edge computing(MEC) service contract formation and workload execution | Ned M.Smith, Beaverton, OR USA; Sanjay Bakshi, Beaverton, OR, USA; Farid Adrangi, Lake Oswego, OR, USA Francesc Guim Bernat, Barcelona, Spain | SLA/Network Service/MEC | H04L 12, H04L 29, G06Q 30 |
| 7 | Control unit and method for the tamper-proof detection of operational safety-related integrity monitoring data | Siemens Aktiengesellschaft. Werner-von-Siemens-StraBe, Augsburg, Germany | Data integrity test | H04W 12, G07C 5, G07C 3 |
| 8 | Wireless network services operating with blockchain technology | Crossover Capital, San Francisco, CA, USA | Exchange cryptocurrency/communication services | G06Q 20, H04L 12, H04W 4, H04W 16, H04B 7, G06Q 20 |
| 9 | System and method for virtual simcard | Donna L. Polehn, Kirkland, WA, USA | Network services | H04W 12, H04W 8 H04B 1, H04W 12 |
| 10 | Method and system for storage and retrieval of blockchain blocks using galois fields | Stephen Lesavich, Kenosha, WI, USA Zachary C.Lesavich, Kenosha, WI, USA | Secure storage and retrieval in P2P/Cloud | G06F 7, G06F 17 |

Patent 1 implements identity authentication for entities entering the wireless communication network, completing the identity registration and verification process through the blockchain, and confirms the registered information in the core network.

Patent 2 uses 5G network slicing technology to achieve virtualization, and uses blockchain to enable network participants to perform authentication, credit audit, collaboration, and information-sharing functions.

Patent 3 describes a collaborative road user safety service that interacts with a coordinating setoff of road users' collaborative safety devices to exchange reliable road safety information. A distributed blockchain is used with the service to coordinate data exchange between collaborative safety device users.

Patent 4 studies the role of blockchain-based 5G network technologies in electrical smart grids and presents a statistical calculation of carbon emissions. There is no underlying

communication technology involved in 5G; thus, this patent is applicable to 5G network communication standards.

Patent 5 uses blockchain technology in 5G networks to provide transmission services through the 5G network. The blockchain technology completes the confidential and safe transmission of vehicle data to realize the control and scheduling in the Internet of Vehicles.

Patent 6 is mainly aimed at providing secure MEC access to 5G edge subsystems, providing multi-functional SLA network services for clients, and dynamically negotiating SLA using a decentralized contract system to support the blockchain platform.

Patent 7 is a tamper-resistant control unit and method for realizing security-related integrity detection data through blockchain. The system supports the 5G communication interface standard.

Patent 8 relates to a system and method related to 5G wireless communication technology, providing a payment system method that uses a wireless communication unit and a wireless base station to provide communication services and allow for the exchange of encrypted currencies. The authorization technology realizes the communication of multiple wireless base stations through the blockchain, such as using wireless devices to implement payment schemes, billing and charging devices, spectrum sharing devices, and electronic money payment protocols.

Patent 9 provides a method for a client device to use a virtual subscriber identity module (vSIM) for client devices, which is equivalent to an identity certificate, making it an authentication mechanism based on the blockchain platform interface.

Patent 10 is designed to retrieve electronic information through a computer network, cloud platform, and peer-to-peer (P2P) application system, using blockchain technology and retrieval methods to solve the problems of privacy and security of electronic content and user retrieval on cloud computer networks.

## 5. Research Significance

### 5.1. Academic Significance for Future Research

The performance of a blockchain system is determined by the underlying communication network, blockchain architecture, consensus algorithms, and distributed applications. Implications of the current study for future research include the following: (1) given the lack of large-scale parallel processing for consensus algorithms, 5G technology can be integrated to provide safe, reliable, and high-performance consensus algorithm research on blockchain; (2) the MEC node and the blockchain network are used as an independent platform to complete their respective functions; (3) covert wireless or optical fiber transmission of blockchain data can be done based on 5G networks; (4) cloud blockchain architectures based on 5G core networks can realize unified communication standardization for edge subsystem MEC in different regions; (5) 5G can be combined with blockchain technology to manage permission data stored in the blockchain of terminal devices connected to 5G networks for information traceability; (6) the characteristics of blockchain technology can be combined to realize edge computing security systems, such as terminal device identity authentication, data security and privacy protection; (7) dynamic spectrum sharing technologies in 5G network-based blockchain smart contracts can be used with intensive networks, with the distributed nature of blockchain and upper-layer smart contracts providing advantages including intelligent settlement, value transfer, and resource sharing; (8) mapping mechanisms for virtual resources can be done using 5G network slicing technology for the blockchain network; and (9) 5G and blockchain technology can be integrated with IoT technology. The blockchain network is deployed in the edge cloud network terminal nodes composed of IoT and 5G, and the IoT terminal devices that are accurately controlled and scheduled through the blockchain and 5G.

### 5.2. Concrete Suggestions for Feasible Implementation

Blockchain transaction performance is mostly determined by the underlying algorithm of the public chain and the level of node hardware. Only when nodes use a large number of

wireless transmission methods for information broadcasting can the concurrent execution of transactions be achieved. Therefore, 5G technology can improve the TPS (Transactions Per Second) performance of the entire blockchain network, thus improving blockchain transactions performance. At this stage, data transmission in most blockchain networks and 5G core networks are still conducted over traditional optical fiber.

The significant effect of 5G technology on the blockchain is to speed up off-chain data transfers in wireless environments. However, accelerating the implementation of 5G in the blockchain field still requires the support of third-party technologies, such as IoT. In addition, many practitioners believe that blockchain can achieve effective data anti-tampering, but it is difficult to ensure the authenticity of the data on the chain. That is, the possibility of artificial tampering remains while the information is on the chain. IoT technologies may help resolve this problem, as replacing manual operations with IoT sensors can increase the cost of data tampering and reduce the risk of false information on the chain.

### *5.3. Implications for Government Policy for Implement Compliance, Standardization, and Governance Control Issues*

(1) Blockchain technology has the potential to enable government agencies and markets to achieve true and transparent project information and dynamic supervision, promoting the transformation of government functions and the innovation of supervision methods, thus effectively improving government service quality. (2) Digital government requires safe and reliable data sharing technology. The distributed, consensual, and encrypted nature of blockchain can help local governments to break through information barriers and open up data silos. (3) 5G networks allow for instantaneous data transmission, giving the public more convenient, efficient, and reliable access to government developments. (4) With the integration of 5G technology and cutting-edge technologies such as blockchain, IoT, artificial intelligence, big data, and virtual reality, governments will adopt more scientific and systematic approaches to policy formulation, supervision, and implementation. (5) Future establishment of core cloud chain platforms in 5G networks will eliminate limitations on government supervision to allow for smooth, barrier-free information supervision.

Achieving this level of government functional efficiency through blockchain not only requires the active exploration of blockchain technology applications but also requires governments to actively assume regulatory responsibility for blockchain technologies and applications. Of critical importance is the establishment of a legal chain, using blockchain technology to supervise the blockchain industry, establishing public trust for such systems, and ensuring regulatory compliance.

### 6. Conclusions

The 5G network is an accelerator for blockchain, and blockchain is a form of body armor for the 5G network. A 5G-based edge subsystem MEC and blockchain network architecture diagram can be constructed. The blockchain can be installed on the MEC platform, but it will increase the burden of edge processing. This raises the need for a blockchain network platform in the edge cloud to process identification, payment transactions, and secure data transmission for the terminal accessing the 5G network.

This paper presents patent analysis of blockchain-based 5G network technology. Research results and limitations are summarized as followed:

1. Search results are taken from the IPETCH patent database as of April 2020. These search results are subject to finite search periods and patent terms. In addition, due to the difference in the search technology of the patent software itself, some but potentially not all unrelated patents will be omitted. The results of patent analysis may feature some deviations, but the mainstream technical direction and research results provided should be of use.
2. Patent life cycle results indicate patents in this domain peaked in 2018, followed by a bottleneck period beginning in 2019. The high cost of chain formation, high

technical thresholds, and high regulatory barriers have slowed the implementation of blockchain technology, thus restricting the integration of the blockchain-based 5G network technology industry.

3. From 2014 to 2020, patent grants were dominated by firms and organizations in five countries: the United States, China, Germany, Finland, and Great Britain. The United States has a particularly strong lead in the combined 5G/blockchain technology domain.

4. The main IPC technologies used by the top seven patent holders are H04L, G06Q, H04W, and G06F.

5. While the deployment feature of blockchain-based 5G network integration architectures is more common, it increases the burden of edge processing, raising the need to establish an independent blockchain network platform in the edge cloud.

6. Patent subject word clustering results were used to generate a technology clustering map. The key applications of blockchain-based 5G network technologies covered a wide number of fields, listed in descending order of importance as follows: computer-implemented methods, smart contracts, distributed ledgers, computing devices, blockchain members, Internet of Things, blockchain authorization, and mobile devices.

7. The relationship between IPC classification technologies was converted into a node-to-node relationship topology graph. The network topology was used to evaluate the correlation and importance between IPCs according to the characteristics of the topology graph nodes. H04L 9, G06Q 20, and H04L 29 were found to be the three key IPC technologies for patent research for the blockchain-based 5G network domain.

This study uses patent analysis methods to investigate development trends and investment opportunities of the blockchain-based 5G network technologies and makes important contributions. Blockchain provides consensus and security assurances, and integrating blockchain into Internet of Things applications will allow for contract transactions to be conducted on the blockchain. Such integration requires the high coverage capabilities of 5G networks. However, such integration still requires considerable effort to achieve technology maturity, stability, and standardization. Implementation of blockchain-based 5G network technologies will allow for identity authentication for intelligent terminal equipment networking, secure data transmission, sharing and transactions, and resource optimization and recycling.

**Author Contributions:** F.G. and M.-H.W. contributed the research concept; F.G. and D.-L.C. collected the data; F.G. and D.-L.C. analyzed the data; F.G. and M.-H.W. wrote the paper; and M.-H.W. and R.-Y.Y. advised the research. All authors have read and agreed to the published version of the manuscript.

**Funding:** This study was funded by the project research on recommendation system based on node multi-source heterogeneous topology data (JT180479). This work was also supported partly by the by project research on the application of artificial intelligence technology in teaching field (JG201806).

**Data Availability Statement:** Data sharing not applicable.

**Conflicts of Interest:** The authors declare no conflict of interest.

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
