# Peer review of "Revealing Development Trends in Blockchain-Based 5G Network Technologies through Patent Analysis"

_sustainability, doi:10.3390/su13052548_

Round 1

Reviewer 1 Report

Dear authors,

I like your research paper a lot. The given study is devoted to a very topical issue of integration blockchain technology and 5G network and I was reading it with high interest. My research interest lies in the field of business digitalization and your research related to patent applications analysis touching also intellectual property rights is highly valuable. 

For a better perception of the research I would recommend improving the quality of Figures 4 and 9.

Good luck,

Faithfully yours,

The Reviewer

Author Response

“For a better perception of the research I would recommend improving the quality of Figures 4 and 9.”

Thanks for the Reviewer 1.  This concern is well received. We have updated the manuscript. We have improving the quality of Figures 4 and 9, and make the Figures clearer. Figures 4 is at the bottom of page 7. Figures 9 is in the middle of page 14. 

Reviewer 2 Report

Overall, the work is interesting. It required a lot of commitment and brings interesting insights.
My doubts are raised while Table 5. Ranking of the R & D capability of patentees.
I don't understand where the individual values came from? Why did you use this formula: R&D = 5θ1 + 2θ2 + θ3 + θ4 - θ5 + 0θ6?

What influences such a system of dependencies of individual parameters?
The description in front of the table does not explain its results and is even confusing / contradictory: "While Alibaba led the field in terms of total number of patent applications and R&D intensity, it’s performance in deep integration of 5G and blockchain was relatively lower." - Relative Capability = 100%
Rows 335-344 - Provide additional explanations, or give up this part of the work because it does not provide relevant information.

Author Response

Reviewer 2,  Concern  1:

“My doubts are raised while Table 5. Ranking of the R & D capability of patentees. I don't understand where the individual values came from? Why did you use this formula: R&D = 5θ1 + 2θ2 + θ3 + θ4 - θ5 + 0θ6? What influences such a system of dependencies of individual parameters?”

        Thanks for the Reviewer 2.  This concern is well received. The parameters corresponding to this formula are the default parameters set in the IPTECH patent database for patent strength evaluation. Then these parameters are supplemented. The added content is highlighted in yellow on page 13 of the paper. The content added is "The greater the number of patent .......... The parameters corresponding to this formula are the default parameters set in the IPTECH patent database for patent strength evaluation.”

Reviewer 2,  Concern  2: 

The description in front of the table does not explain its results and is even confusing / contradictory: "While Alibaba led the field in terms of total number of patent applications and R&D intensity, it’s performance in deep integration of 5G and blockchain was relatively lower." - Relative Capability = 100%. Rows 335-344 - Provide additional explanations, or give up this part of the work because it does not provide relevant information.”

         Thanks for the Reviewer 2.  This concern is well received. The added content is highlighted in red on page 13, it mainly describes the reason why Alibaba led the field in terms of total number of patent applications and R&D intensity, it’s performance in deep integration of 5G and blockchain was relatively weak. The content added is “While Alibaba led the field in terms of .......... in terms of the degree of deep integration of 5G and blockchain technology.”

Reviewer 3 Report

In this paper, the authors study how 5G network technologies make it possible to propose a network architecture using mobile edge computing (MEC) and blockchain as in-dependent platform components to solve MEC load pressure.

The preliminary study of the integration of blockchain and MEC or edge computing technology is adequate and includes sufficient sources of information. The selected references are adequate, correspond to the state-of-the-art Works, and belong to high quality conferences and journals.

In Section 4.1 I miss some more analysis in relation to the provision of technologies related to 5G, such as the emerging 6G, and other terms related to 5G, such as the air interface defined by 3GPP for 5G (known as New Radio, NR) , or “network slicing”, which is the network architecture that enables the multiplexing of virtualized and independent logical networks on the same physical network infrastructure.

It would be interesting to see if these terms directly related to 5G appear frequently in patents alongside blockchain.

Overall, I think the paper is well written and the analysis carried out is accurate.

Some corrections:

Lines 88 to 95:  Substitute “The first part” “The second part”, etc, for “Section 2”, “Section 3”.

Author Response

“Lines 88 to 95:  Substitute “The first part” “The second part”, etc, for “Section 2”, “Section 3”.”

Thanks for the Reviewer 3.  This concern is well received. In the last paragraph of the body content of page 2, the yellow font for highlighted changes is the revised content. (Lines 88 to 95).